# Vibration of Natural Rock Arches and Towers Excited by Helicopter-Sourced Infrasound

Riley Finnegan[1], Jeffrey R. Moore[1], Paul R. Geimer[1]

[1]Department of Geology and Geophysics, University of Utah, Salt Lake City, 84112, USA

*Correspondence to: Riley Finnegan (riley.finnegan@utah.edu)*

**Abstract.** Helicopters emit high-power infrasound in a frequency range that can coincide with the natural frequencies of rock landforms. While a single previous study demonstrated that close-proximity helicopter flight was able to excite potentially damaging vibration of rock pinnacles, the effects on a broader range of landforms remain unknown. We performed a series of controlled flights at seven sandstone arches and towers in Utah, USA, recording their vibration

response to helicopter-sourced infrasound. We found that landform vibration velocities increased by a factor of up to 1000 during close-proximity helicopter flight as compared to ambient conditions immediately prior, and that precise spectral alignment between infrasound and landform natural frequencies is required to excite resonance. We define admittance as the ratio of vibration velocity to infrasound pressure and recorded values up to 0.11 mm s$^{-1}$ Pa$^{-1}$. While our results demonstrate a strong vibration response, the measured velocities are lower than likely instantaneously damaging values. Our results serve

as a basis for predicting unfavorable degradation of culturally significant rock landforms due to regular helicopter overflights.

## 1 Introduction

Anthropogenic activity is increasingly shaping evolution of the natural environment over geologically brief timescales (Aarons et al., 2016; Cochran et al, 2013; Whyte, 2016). Motor vehicles, trains, and aircraft generate ground- and air-borne

vibration energy that may propagate over large distances, significantly altering the ambient vibration wavefield in some regions (Riahi and Gerstoft, 2015). Helicopters, however, are unique in that they have nearly limitless reach over the landscape and produce a range of sound energy during operation. Helicopters create "thickness noise," directional, narrow-band infrasound emitted from the main rotor blades and strongest in the rotor plane (Schmitz and Yu, 1983), which while inaudible, is often the loudest sound produced and propagates over large distances (Cheremisinoff, 1994). This energy

occurs in a narrow frequency band, the so-called blade pass frequency (and its overtones), which is set by the number of main rotor blades multiplied by the rotor revolution rate, and the resulting measured sound frequency is only subsequently altered by Doppler shift during flight (Doppler, 1842; Schmitz and Yu, 1983). These helicopter and other acoustic signals couple to the ground (Bass et al., 1980) and can be recorded as Raleigh waves using seismometers (Novoselov et al., 2020).

The corresponding seismic signals resemble tremors produced by volcanoes (Eibl et al., 2015) and researchers have used seismic measurements of helicopter-sourced infrasound to investigate flightpaths near airfields (Meng and Ben-Zion, 2018), volcanoes (Eibl et al., 2017 and Eibl et al., 2015), and glaciers (Podolskiy et al., 2017).

Past studies have assessed the effects of helicopter-sourced infrasound on people and engineered structures (Broner, 1978; Hanson et al., 1991; Schomer and Neathammer, 1985; Wagner, 1978); however, few have analyzed the effects of directional infrasound generated by thickness noise on the vibration response of natural landforms and other cultural features (King, 1996, 2001). A single previous study (King, 2001) reported that a helicopter hovering 10 m from a rock pinnacle was capable of exciting vibrations with measured peak velocity of several mm s$^{-1}$, which is in the range considered as potentially damaging for geologic features and ancient monuments (Whiffin and Leonard, 1971; Hanson et al., 1991; Hendricks, 2002; Volpe, 2014; Moore, 2018). When the frequency of an energy source – in this case, helicopter-sourced infrasound – precisely matches the fundamental or higher-order natural frequencies of a structure, resonance is excited and vibration amplitudes increase markedly, potentially generating damage in the structure (Fujino et al., 1993). Geologic structures such as natural arches and towers are abundant in the southwestern United States and are also found in other environments around the world. As helicopter flights are increasingly common in these areas (National Park Service 2016, 2020), additional studies are needed to characterize the effects of helicopter activity, especially sustained activity over many years, on structural degradation of these valued landforms.

Here, we explore the vibration response of natural sandstone arches and towers (or hoodoos) to helicopter-sourced infrasound. We performed a suite of field measurements where we recorded landform vibration response during controlled helicopter flights. We assessed landforms of different sizes and geometries and varied helicopter maneuvers. We found that infrasound emitted from helicopters can excite resonance of natural landforms, albeit at peak velocities generally lower than those thought to produce instantaneous damage. Our results have implications for the management, conservation and preservation of culturally valuable rock landforms.

## 2 Study Sites

We selected 4 arches and 3 towers for study during helicopter flights, all in Utah, USA (Fig. 1, Table 1). These landforms are on the traditional lands of the Hopi, Navajo, Southern Paiute, Ute, and Zuni peoples, and similar arches and towers have cultural significance to these populations (Stoffle et al., 2016). Additionally, many arches and towers serve as tourist attractions in the desert southwest and see millions of visitors each year (National Park Service, 2021). The selected arches have spans of between 3 and 12 m and the hoodoos are 4 m tall (Table 1). Arsenic Arch and Squint Arch are formed in

Navajo Sandstone of the Glen Canyon Group, Big Arrowhead Arch is formed in Cedar Mesa Sandstone of the Cutler Formation, and Two Bridge is formed in sandstone of the Pink Member of the Claron Formation. The hoodoos are part of the Little Egypt outcrop formed in Entrada Sandstone.

## 3 Methods

We first identified landforms that would likely be susceptible to infrasound emitted by common civilian helicopters. Using "infraBSU" infrasound microphones (Marcillo et al., 2012), we recorded the blade pass and higher-overtone frequencies (to which we collectively refer as infrasound here for simplicity) of thickness noise from a 2-blade Bell 206 helicopter. We additionally recorded 3- and 5-blade helicopters to verify the frequency of infrasound blade pass frequency of these emitted by models according to the different number of main rotor blades (Table 2). Using Nanometrics Trillium Compact 20-s seismometers and Fairfield Zland 5-Hz nodal geophones, we recorded ambient vibration of 10 arches and 10 towers by temporarily deploying an instrument directly on each landform, leveled and aligned to magnetic north. The instruments recorded vibrations on average for 1–2 hours with sampling rates between 100 Hz and 1 kHz. Following procedures outlined by Starr et al. (2016) and Moore et al. (2018), and using methods described by Koper and Burlacu (2015) and Koper and Hawley (2010), we created power spectral density (PSD) estimates for each landform to determine the natural frequencies of the arches and towers from ambient seismic recordings (Fig. 1). We then calculated frequency-dependent polarization attributes to obtain the azimuth, incidence, and degree of polarization for each identified natural frequency (Figs. A1–A7; IRIS DMC, 2015). We confirmed the field results with numerical eigenfrequency modeling, following methods of Geimer et al. (2020). We acquired ground- and aerial-based photos to construct photogrammetry models of the landforms using Agisoft PhotoScan Professional (now Agisoft Metashape, www.agisoft.com) and Bentley ContextCapture (bentley.com), and imported these models into the finite-element simulation software COMSOL Multiphysics (www.comsol.com). We assigned uniform material properties of density, Young's modulus, and Poisson's ratio, along with boundary conditions to each model, and adjusted Young's modulus until we produced a best match to the field data. While numerical modal analysis was not integral to this study, our data provide only a limited description of modal displacement for each landform while numerical models provide a detailed approximation of the modal deformation field (Fig. B1).. From the initial 20 landforms assessed, we selected the 4 arches and 3 towers with natural frequencies near 13 Hz or 26 Hz to study their vibration response during controlled flight of a Bell 206 helicopter.

We chartered several helicopter flights where we measured vibration of the arches and towers along with nearby reference sites, infrasound levels, and the helicopter's position (Fig. 2a–b). Nodal geophones and broadband seismometers sampled data between 100 Hz and 1 kHz, while the infraBSU microphones sampled at 200 Hz using a 24-bit DataCube data logger.

At Two Bridge (Bryce Canyon National Park, UT), the helicopter flew a standard 35-minute tourism flight over the park, with infrasound reaching the bridge at observable levels for ~4 minutes (Fig. 2c–f), while at the other sites, the helicopter performed specified maneuvers designed to vary the helicopter's speed, distance, angle, and direction for 11–25 minutes. We used a handheld GPS to measure the helicopter's position, which at Two Bridge sampled every second, while at the other sites the sampling rate varied in relation to helicopter speed (i.e. higher sampling rates at faster speeds). In processing landform vibration data, we removed each instrument's response to work in consistent units of vibration velocity and velocity-derived vibration power. The infraBSU microphones have a flat response over the 0.038 Hz corner frequency and a sensitivity of 4.5E-5 V Pa$^{-1}$; we multiply the data by a factor of 8.744E-5 Pa per count to convert raw data to pascals. To compare the vibration and infrasound records, we resampled each time series to a common sampling frequency, and then linearly interpolated the helicopter position data.

We use admittance as a measure of a landform's vibration susceptibility to helicopter-sourced infrasound. Admittance is formally defined as the ratio of a velocity vector to a force vector (Olesen and Randall, 1977), measured at the same point, as a function of frequency. Here, we represent admittance as the linearized scaling relationship between vibration velocity and infrasound pressure, considering magnitude only and not direction, to describe a landform's vibration velocity response to infrasound pressure. Greater admittance values thus indicate that higher vibration velocities are excited for a given infrasound input. To calculate this value, we first selected the orientation of maximum vibration for each landform: for arches, we used the vertical component of velocity, per previous modal analyses (Geimer et al., 2020), while for hoodoos, we rotated the horizontal data to the dominant polarization azimuth at the natural frequency of interest (Figs. A1–A7). We filtered infrasound and vibration data between 10 and 17 Hz or 22 and 30 Hz to isolate frequencies corresponding to the landform's natural frequencies and the helicopter's blade pass frequency and first overtone (i.e., 13 and 26 Hz). We plotted the smoothed root-mean-square (RMS) envelope of the absolute value of arch vibration velocity as a function of the smoothed RMS envelope of the absolute value of infrasound pressure, in 10-s increments with 90% overlap, for each maneuver during the helicopter flights (Fig. 3f). We then fit these increments of data with a linear relationship and took the slope as admittance. We finally averaged all admittance values calculated for each individual helicopter maneuver.

We also used the Doppler equation to predict the shift of infrasound frequency from the helicopter's motion (Appendix C). We obtain parameters like closest distance to arch, speed of helicopter, and time at closest distance to arch from the GPS track generated onboard to predict the shifted infrasound frequencies. We used this frequency range and our 3D arch models to model arch vibration response with different energy source directions.

## 4 Results

We recorded infrasound from different helicopters and employed a 2-blade Bell 206 helicopter for controlled-flight field measurements. Our data confirmed that helicopter-sourced infrasound occurs at frequencies corresponding to a helicopter's blade pass frequency and integer-multiple overtones (Fig. 1a, Table 2). Our data also showed that the infrasound pressure at an observer depends on the helicopter's distance and wind conditions. We recorded peak infrasound pressure of 11.2 Pa, which corresponds to a sound pressure level of 115.0 dB. In our tests, helicopter infrasound pressure decreased with increasing distance (Fig. D1), as anticipated for a point source of sound.

We additionally used parameters from the helicopter's GPS track to trace the infrasound frequency shift during straight-line flight. We found the Bell 206 infrasound frequencies shifted by ±15–20% from the stationary blade pass frequency and overtones, i.e., 13 Hz energy shifted between 11.1 Hz and 15.6 Hz, and 26 Hz energy varied from 22.2 Hz to 31.3 Hz (Appendix C, Fig. C1). Other helicopter models with different speed capabilities could produce greater Doppler shift.

Arch and tower vibration velocities increased 100–1000 times during helicopter flight as compared to ambient conditions immediately prior (Figs. 2e and 3b). Maximum landform vibration amplitude during helicopter flight was between 0.007–0.13 mm $s^{-1}$. Maximum admittance (slope of the best fit line between landform vibration velocity and helicopter infrasound pressure) ranged between 0.01–0.11 mm $s^{-1}$ $Pa^{-1}$ (Figs. 3 and 4). We found admittance magnitude was predominantly affected by frequency alignment between helicopter-sourced infrasound and landform modal frequencies, along with distance between the helicopter and landform. Admittance was secondarily affected by landform geometry, alignment between landform modal vectors and infrasound propagation direction, and the propagation of infrasound itself.

## 5 Discussion

### 5.1 Primary factors controlling landform response

Our data show that precise spectral alignment between helicopter-sourced infrasound frequencies and a landform's natural frequencies is one of the primary contributing factors to a landform's vibration response (Figs. 2c–d and 3a–c). This includes any combinations between the potentially Doppler-shifted blade pass frequency and higher overtones of helicopter-sourced infrasound, and fundamental or higher-order natural frequencies of the landforms.

Vibration velocity increased at all sites where Doppler-shifted infrasound frequencies aligned with the landform's natural
frequencies. This demonstrates that even while infrasound pressure increases at closer distances, alignment between source and landform frequency plays a crucial role in controlling the landform response. Our field experiments included straight-line flyovers, as well as turns. During both types of maneuvers, infrasound frequencies were Doppler shifted, and when these precisely aligned with the landform's natural frequencies, vibration velocity increased markedly. Tracking admittance over time during the helicopter approach at Arsenic Arch, we observed a strong increase in admittance as Doppler-shifted
infrasound progressively aligned with the landform natural frequency (Fig. 3). In turn, spectral alignment between Squint Arch's 13.8 Hz natural frequency and Doppler-shifted infrasound when the helicopter was 2000 m away from the arch caused the arch to vibrate at the same velocity as when the helicopter was 350 m away but with poorer alignment between natural and infrasound frequencies. Two Bridge, which resonates at 13.8 Hz, achieved peak vibration power slightly before the helicopter infrasound reached peak sound pressure level (Fig. 2f), again due to Doppler shift aligning the helicopter blade
pass frequency with the bridge's fundamental frequency. We attribute the necessary precision of frequency alignment observed in our results to the relatively low modal damping ratios of these landforms (1–2%; Geimer et al., 2020), as low damping results in narrower spectral peaks and thus a narrow bandwidth of sensitivity to input energy.

Landforms like Arsenic Arch and the Little Egypt hoodoos, which have natural frequencies around 26 Hz, had highest admittance values in the 22–30 Hz energy band. These landforms do not have natural frequencies at 13 Hz and their
respective admittance values in the 10–17 Hz band were negligible (despite high infrasound pressures in this band). Conversely, Two Bridge and Big Arrowhead Arch do not have 26 Hz natural frequencies, and admittance in the 22–30 Hz band was minor compared to admittance in the 10–17 Hz band. Squint Arch resonates at 12.5, 13.8 and 26.0 Hz, and thus had nonzero admittance in both frequency ranges.

Instrument placement is vital for measuring a landform's maximum admittance: measuring vibration velocity at the point of
peak modal deflection provides an accurate assessment of the largest induced vibration response. Squint Arch's 26.0 Hz natural frequency represents a second-order bending mode, and the sensor placed at the 'A' location on the arch is at a node point (Fig. 4b). The recorded admittance at 'A' in the 22–30 Hz band was thus almost zero, but was highest in the 10–17 Hz band where location 'A' corresponds to maximum deflection of the fundamental mode. Results from Squint Arch also demonstrate that higher helicopter infrasound pressure emitted at 13 Hz versus 26 Hz, as well as higher arch vibration
velocity at a landform's fundamental frequency versus higher-order frequencies, result in greater admittance.

Admittance was largest for close-proximity, low-speed helicopter approaches where landform and infrasound frequencies aligned closely and Doppler shift was minor (Fig. 4a, Table E1). However, at Big Arrowhead Arch, which has two natural frequencies on the very low and high bounds of Doppler-shifted helicopter infrasound, low-elevation hovers produced small admittance values. Instead, variable-speed circular maneuvers around the arch resulted in large Doppler shift of infrasound and higher admittance than during non-Doppler-shifted periods. The tourist helicopter flight above Two Bridge demonstrated that while close-proximity flight produces the highest landform vibration response, helicopter infrasound propagates over large distances affecting landforms over a large radius. The helicopter's closest distance to the bridge was 600 m, yet the vibration velocity increased by a factor of more than 100 times during the flight compared to conditions immediately prior (Fig. 2e).

## 5.2 Secondary influences on landform response

Additional factors affecting landform vibration response include alignment of modal vectors with the direction of infrasound propagation, landform geometry, and environmental influences. As infrasound pressure is greatest in the plane of the helicopter's main rotor, the direction of infrasound propagation is predominantly horizontal during level flight, and precise horizontal alignment with landform modal vectors increases admittance. Helicopter approaches and hovers at the hoodoos, which have bending modes similar to that of a vertically-oriented cantilever, resulted in the highest admittance. This implies that if a helicopter maneuvered in such a way that its main rotor is perpendicular to the ground, which is uncommon, landforms with vertically-polarized modes could have higher admittance values comparable to the hoodoos. We tested this using COMSOL Multiphysics, where we aligned the direction of an applied periodic force to the modal vector of a synthetic arch model, and found the arch's vibration velocity increased 7–8 times compared to the case of poor alignment.

We attribute the relatively low admittance measured at Squint Arch to the split mode at 12.5 and 13.8 Hz (Geimer et. al, 2020). The modal vectors at these frequencies have similar azimuth but 60° difference in incidence angle, which may result in partial destructive interference, reducing the arch's vibration response to helicopter flight. We explain the relatively low admittance at Egypt 4 by considering the landform's aspect ratio (Table 1). This hoodoo is much wider than its counterparts, which contributes to greater stiffness and a reduced vibration response. Environmental factors such as topography and wind additionally affect infrasound propagation (Cheremisinoff, 1994), which can cause variations in admittance. High winds produced by helicopter downwash during landing or hovering distort the admittance, as pressure from wind is spread across a broad frequency range. We thus excluded times of windy conditions and/or excessive helicopter downwash from

admittance calculations. As the helicopter settled into its hover position and wind reduced, admittance returned to a stable level (Fig. 3c).

### 5.3 Significance of excitation

Our study measured landform vibration velocities up to 0.13 mm s$^{-1}$, which is an order of magnitude below levels often considered potentially damaging for ancient and culturally valuable structures as well as geologic features (Whiffin and Leonard, 1971; Hanson et al., 1991; Hendricks, 2002; Andrews et al., 2013; Volpe, 2014; Moore, 2018). The measured admittance values for these landforms, however, suggest that given higher infrasound pressure, it would be possible for some of the features to experience potentially damaging vibration velocities. The highest admittance we recorded was 0.11 mm s$^{-1}$ Pa$^{-1}$ at the Egypt 2 hoodoo, and the highest infrasound pressure we recorded was 11.2 Pa during the Squint Arch flight: in a worst-case scenario, certain landforms could experience vibration velocities up to 1.2 mm s$^{-1}$. This velocity could likely be reached for a slender tower or hoodoo with a fundamental frequency at 13 Hz (or the respective blade pass frequency of other helicopter models), and with the helicopter hovering or approaching slowly at a similar elevation as the landform. A past study reported peak vibration velocities of 4.1 mm s$^{-1}$ for a ~13 m tall rock pinnacle (with similar values for two neighboring pinnacles), demonstrating the viability of these elevated amplitudes (King, 2001). We note that for a flat lying fixed-fixed beam analogous to Squint Arch and excited at its fundamental mode, analytical theory indicates this vibration velocity corresponds to an additional longitudinal stress of 12 kPa. To evaluate the impact of this stress on a rock landform, however, requires acknowledging the role of preexisting cracks and a shift into a fracture mechanics framework. While the added stress is relatively small, even minor additional loads can contribute to a large increase in subcritical crack growth rates (Eppes and Keanini, 2017), and thus have strong implications for long term structural health. We additionally note that heavier military helicopters not studied here likely generate higher power infrasound than lighter civilian models (Hanson et al., 1991), and further study is needed to test the effects of military helicopter overflights on the vibration response of rock landforms.

While conditions for exciting high-amplitude resonance in a landform with a 13 Hz fundamental frequency is comparably narrow, the range of features susceptible to excitation by helicopter-sourced infrasound remains broad. There are many natural rock arches and towers, including features documented by Geimer et al. (2020) and Starr et al. (2015), with fundamental and higher order natural frequencies within the frequency range of Doppler shifted infrasound. Similar to how higher order overtones excited resonance at Arsenic Arch's fourth mode, the second and third modes of Moonrise Arch, along with the fifth modes of Mesa, Aqueduct, Sunset, and Double O Arches all have the potential to be excited by

helicopter-sourced infrasound. Additionally, there are well-known areas with clusters of hundreds of rock towers and hoodoos, such as Bryce Canyon National Park (USA), Goblin Valley State Park (USA), Pinnacles National Park (USA), the Djavolja varos (Serbia), and Putangirua Pinnacles (New Zealand), where there is a high likelihood for many landforms to have natural frequencies that coincide with Doppler shifted helicopter-sourced infrasound and thus be susceptible to excitation by helicopter flight activity.

## 6 Conclusion

We performed a series of controlled helicopter flights near several sandstone arches and towers in Utah, recording the vibration response of these landforms to helicopter-sourced infrasound. We found that landform vibration velocities were up to 1000 times greater during helicopter flight than during prior ambient conditions, with peak velocities reaching 0.13 mm $s^{-1}$. We used the maximum measured admittance (a measure of landform vibration susceptibility to helicopter-sourced infrasound) of 0.11 mm $s^{-1}$ $Pa^{-1}$ and maximum recorded infrasound pressure of 11.2 Pa to present a scenario under which a landform could experience potentially harmful vibration during helicopter flight. While this scenario is limited, there are thousands of arches, towers, and other culturally valuable rock landforms in Utah alone (e.g., Stevens and McCarrick, 1988), and a multitude of other nationally significant geologic features around the world. These landforms can experience helicopter overflights up to hundreds of times per day (e.g., National Park Service, 2020). Our study demonstrates that helicopter blade pass frequency and higher-octave infrasound can excite resonant vibrations in rock landforms, and that landforms can be affected by helicopter-sourced infrasound when natural frequencies align with the Doppler-shifted infrasound frequency range. This recurring exposure to helicopter sound energy has implications for the landforms' structural health; our study thus provides the basis upon which further investigation into the degradation of rock landforms due to repeat exposure can be conducted.

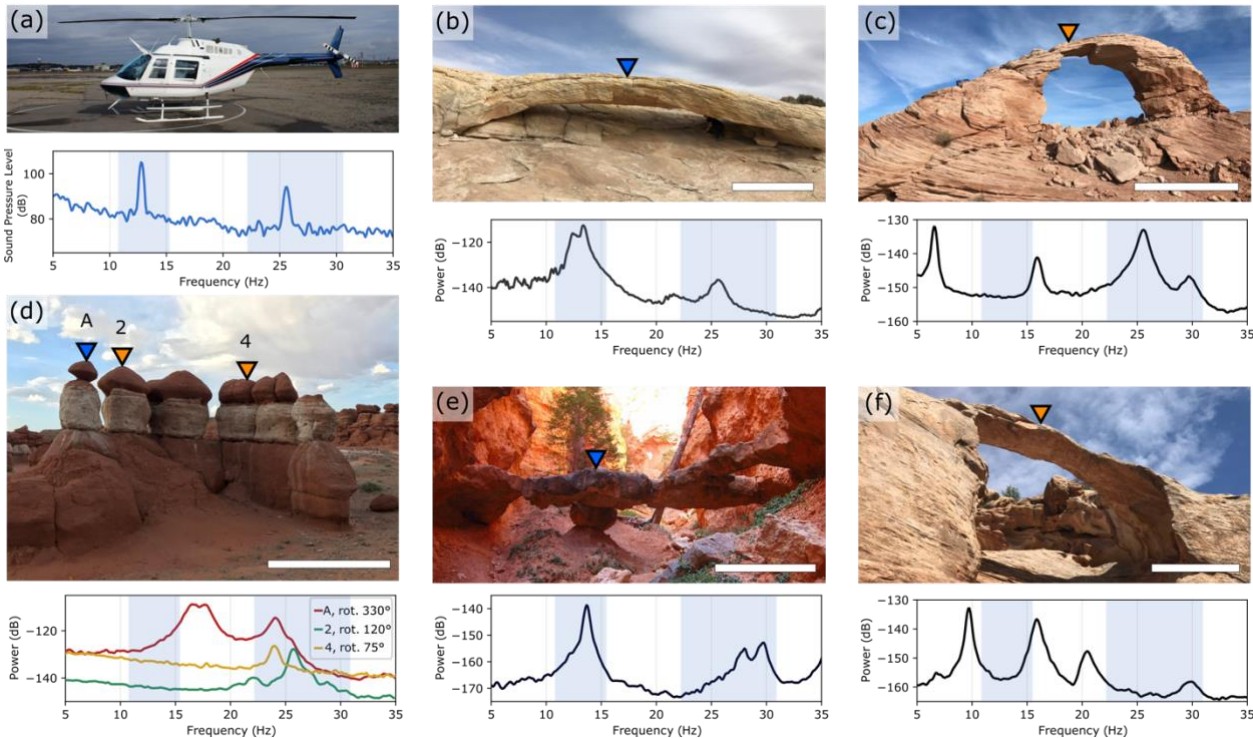

**Figure 1** (a) Bell 206 helicopter (credit: Jan Ainali) and stationary infrasound frequency spectrum, with decibel amplitudes relative to 20E-6 Pa Hz$^{-1}$. (b) Squint Arch, (c) Arsenic Arch, (d) Little Egypt, (e) Two Bridge, (f) Big Arrowhead Arch. White bars are approximately 3 m. Velocity spectra for the landforms are shown below each photo, with decibel powers relative to 1 m$^2$ s$^{-2}$ Hz$^{-1}$. Spectra for arches show the vertical component, while spectra for hoodoos are rotated to the polarization azimuth for the respective mode. The blue shaded region shows the Doppler shifted range of helicopter-sourced infrasound (11.1–15.6 Hz and 22.2–31.3 Hz). Blue triangles indicate locations of broadband seismometers, orange triangles show locations of nodal geophones.


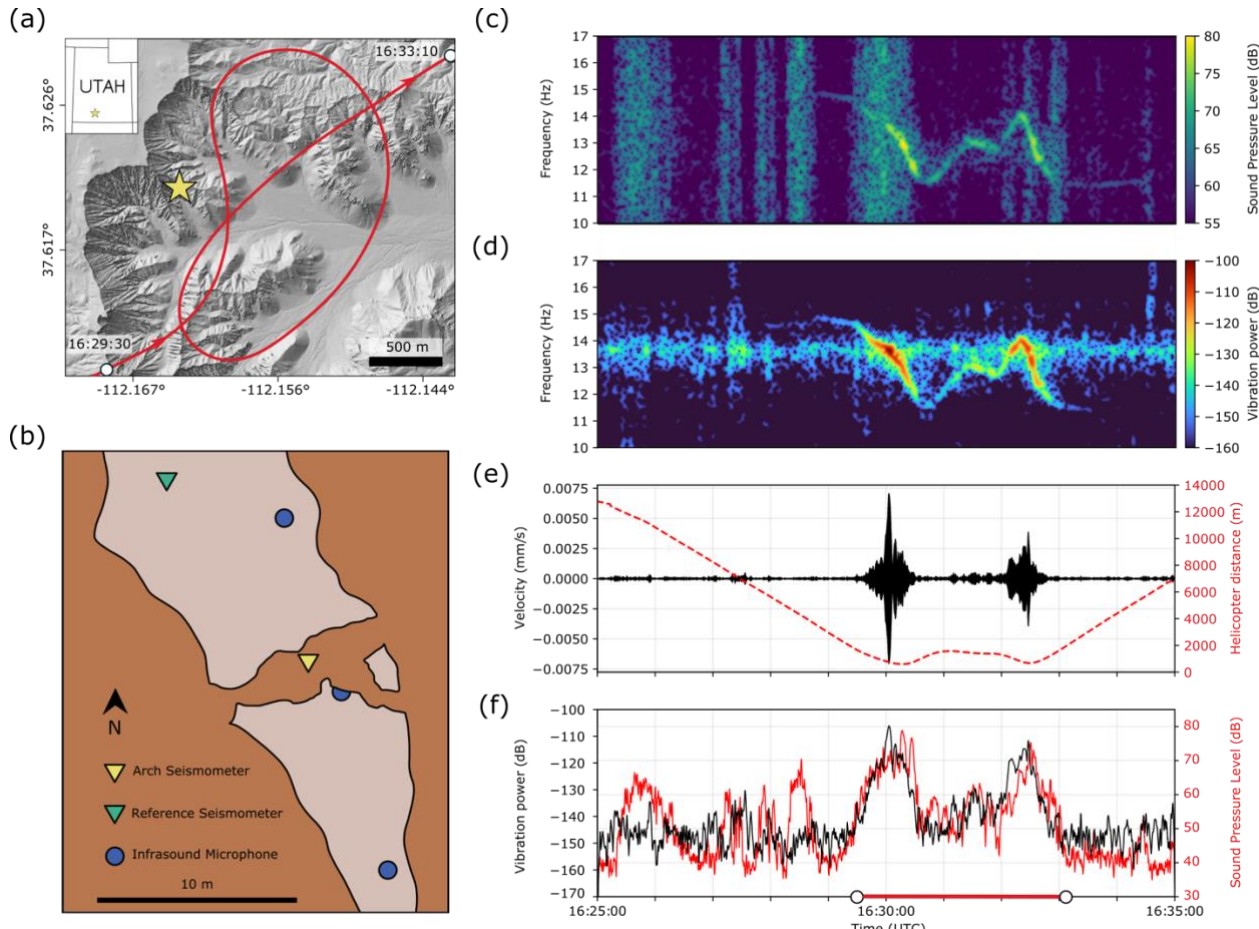

**Figure 2** (a) GPS track of the tourist helicopter flight over Bryce Canyon National Park: the helicopter arrives from the south; Two Bridge location is indicated by the yellow star. (b) Overhead schematic of field equipment layout. (c) Infrasound sound pressure level spectrogram. (d) Two Bridge vibration velocity spectrogram. e) Two Bridge vibration velocity (black) and helicopter distance from the bridge (red) over time. (f) Arch vibration power (black) and infrasound sound pressure level (red). White circles on lower axis correspond to start and end times of the helicopter's flyover in (a). Decibel amplitudes in (c) and (f) are relative to 20E-6 Pa Hz$^{-1}$ for sound pressure level, while decibel powers in (d) and (f) are relative to 1 m$^2$ s$^{-2}$ Hz$^{-1}$ for vibration power.

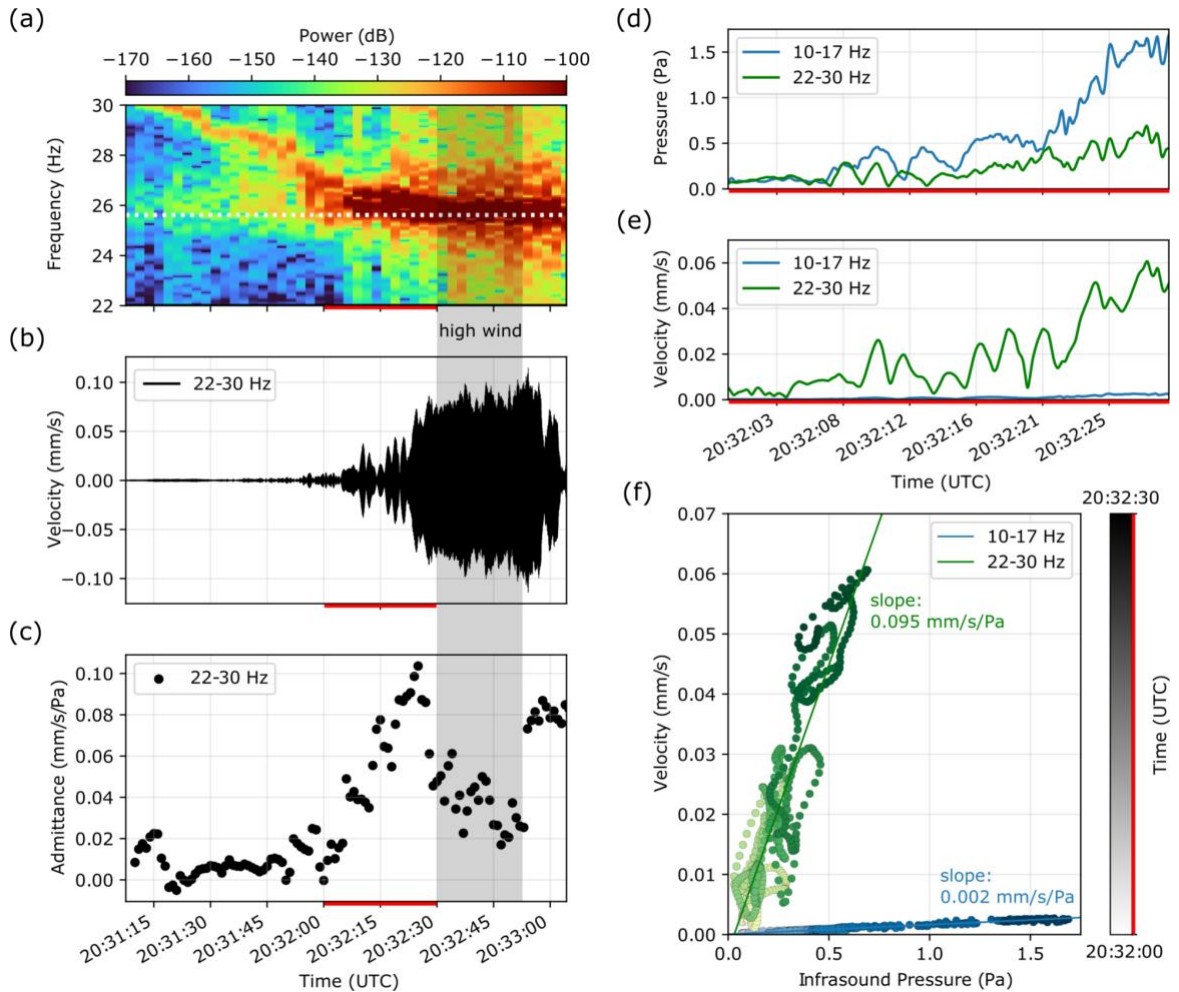

**Figure 3** (a) Vibration velocity spectrogram of Arsenic Arch during helicopter approach, with decibel powers relative to 1 $m^2 s^{-2} Hz^{-1}$. Dashed line highlights the arch's 25.6 Hz natural frequency, which aligns with Doppler-shifted infrasound. (b) Arsenic Arch vibration velocity. (c) Arsenic Arch admittance plotted for 10 s windows with 90% overlap. Gray shaded window in (a–c) is during high winds caused by downwash. (d) Absolute value smoothed RMS envelope of helicopter infrasound pressure. (e) Absolute value smoothed RMS envelope of vibration velocity. (f) Linear fit between arch vibration velocity and helicopter infrasound pressure, where the resulting slope is admittance. Bold red line on time axes in (a–f) indicate same time period.

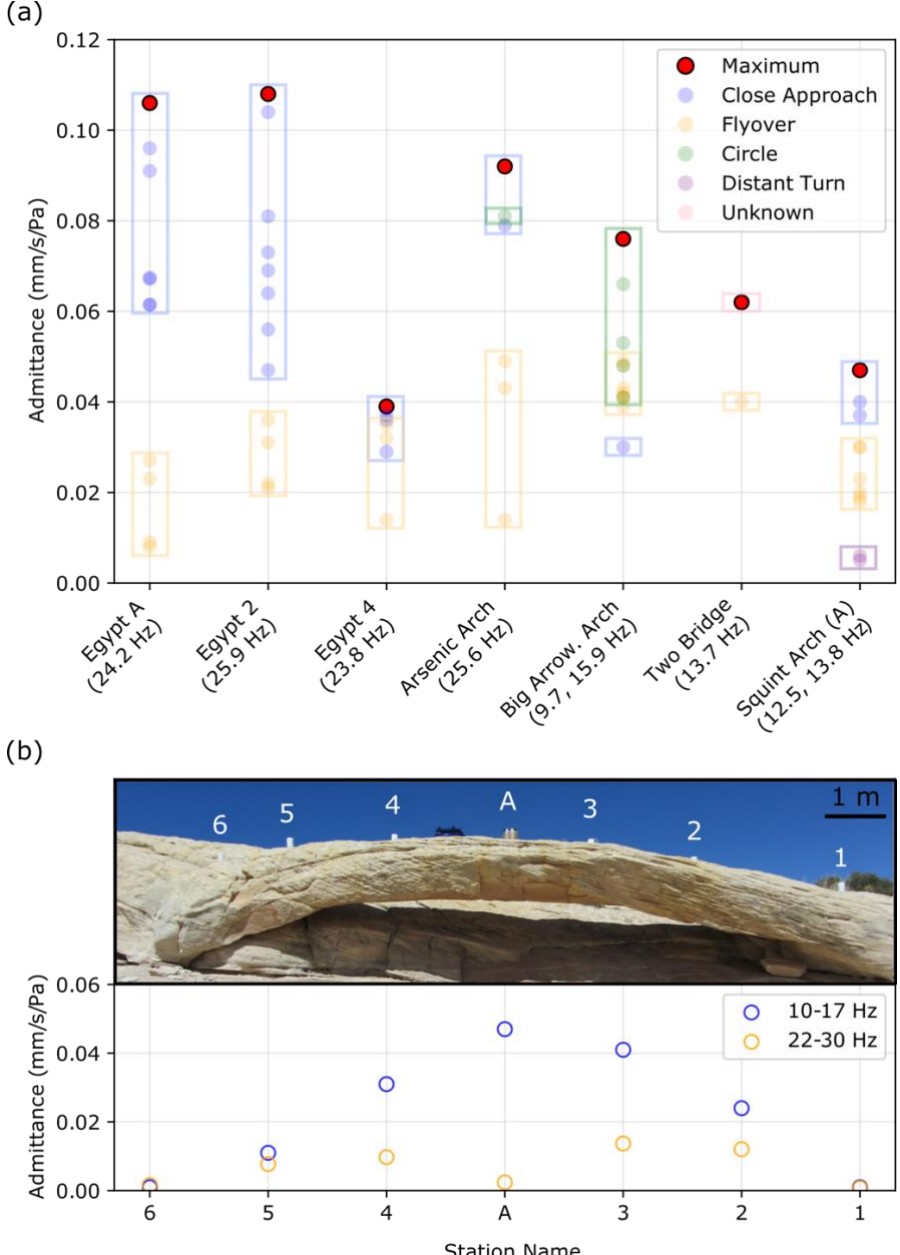

**Figure 4** (a) Admittance values for different maneuvers at all study sites. Landform natural frequency excited is noted in
parentheses beneath each label. (b) Maximum admittance values at different locations on Squint Arch. Numbers (nodal
geophones) and letter (broadband seismometer) above the arch indicate station location.

**Table 1** Study site characteristics, including location, natural frequencies of interest, dimensions, and Young's modulus of the rock mass (E). Dimension A is the longest or tallest axis of each landform, B is the second largest, and C is the smallest. Note these axes are roughly mutually perpendicular.

| Site name | Location Lat., Lon. (decimal degrees) | Natural Frequency (Hz) | A (m) | B (m) | C (m) | E (GPa) |
|---|---|---|---|---|---|---|
| Arsenic Arch | 38.1056, -110.5391 | 25.6 | 3 | 2 | 1 | 0.8 |
| Big Arrowhead Arch | 37.7396, -110.2708 | 9.7, 15.9 | 7 | 3 | 1 | 2.7 |
| Little Egypt 2 | 38.0783, -110.6283 | 25.9 | 4 | 2 | 2 | 1.0 |
| Little Egypt 4 | 38.0783, -110.6283 | 23.8 | 4 | 3 | 2 | 1.0 |
| Little Egypt A | 38.0783, -110.6283 | 24.2 | 4 | 1 | 1 | 1.0 |
| Squint Arch | 38.6465, -110.6739 | 12.5, 13.8, 26.0 | 12 | 2 | 1 | 2.1 |
| Two Bridge | 37.6212, -112.1639 | 13.7 | 9 | 3 | 1 | 5.2 |


**Table 2** Helicopter models, number of main-rotor blades, blade pass frequency ($f_0$) and first overtone ($f_1$) of helicopter-sourced infrasound and sound. Data are compiled from measurements made by the research team (indicated by an asterisk) and from available literature (see Moore, 2018). $f_1$ not measured for Kaman K-1200 model.

| Helicopter Model | Number of Blades | $f_0$ (Hz) | $f_1$ (Hz) |
|---|---|---|---|
| Kaman K-1200* | 4 (2 sets of 2) | 9 | -- |
| Bell UH-1 | 2 | 11 | 22 |
| Bell 206* | 2 | 13 | 26 |
| Robinson R44* | 2 | 13 | 26 |
| Eurocopter AS350/Airbus H125* | 3 | 20 | 40 |
| Eurocopter EC130/Airbus H130 | 3 | 20 | 40 |
| Bell 407* | 4 | 27 | 54 |
| MBB Bo 105* | 4 | 28 | 56 |
| MD 500* | 5 | 40 | 80 |


## Appendix A: Spectral and Polarization Attributes of Landforms

Figures A1–A7 show spectral and polarization attributes of the landforms assessed in this study. We processed ~2 hours of ambient vibration data for each arch or hoodoo, filtering the data between 0.1 and 40 Hz and removing the instrument response. We created acceleration power spectral density (PSD) plots for each unrotated component (vertical, east-west, and magnetic north-south), and computed polarization attributes of azimuth, incidence, and degree of polarization for 600-second windows subdivided into 20 segments with 50% overlap. We show probability densities and the median for each.

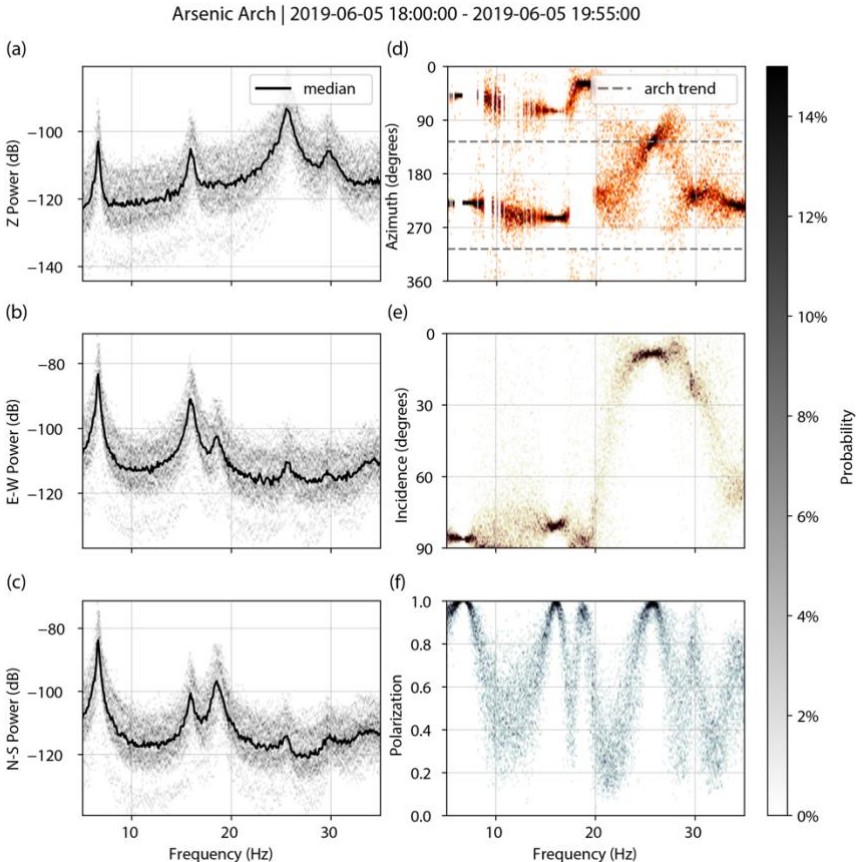

**Figure A1.** Arsenic Arch. PSDs for vertical (a), east-west (b), and north-south (c) components. Decibel powers are relative to 1 $m^2$ $s^{-4}$ $Hz^{-1}$. Polarization attributes of azimuth (d), incidence (e), and degree of polarization (f).

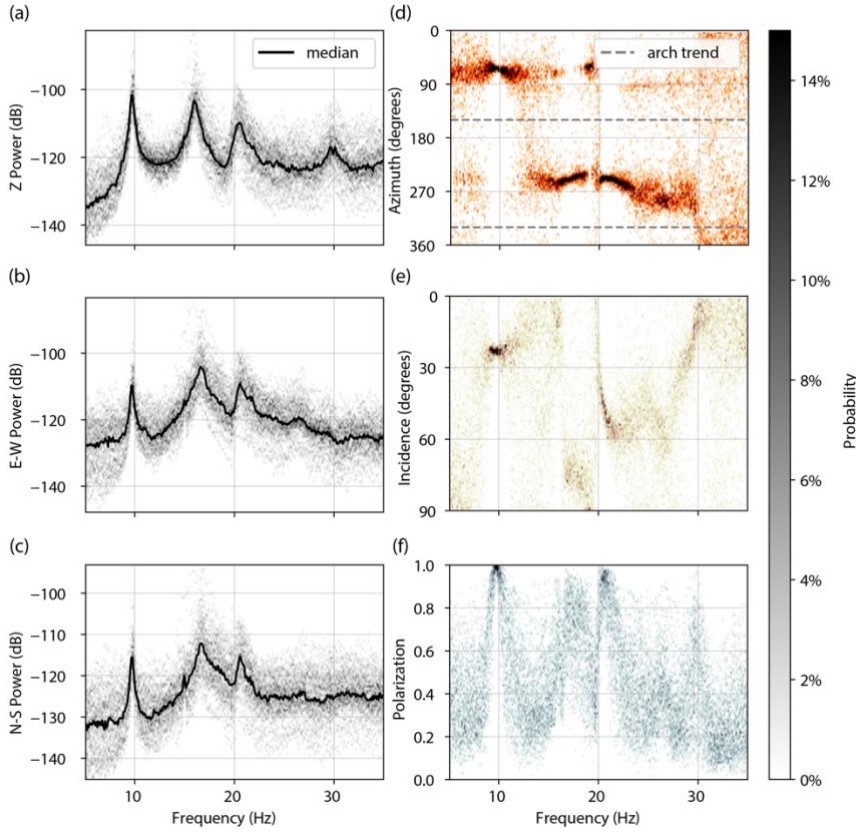


**Figure A2.** Big Arrowhead Arch. PSDs for vertical (a), east-west (b), and north-south (c) components. Decibel powers are relative to 1 $m^2$ $s^{-4}$ $Hz^{-1}$. Polarization attributes of azimuth (d), incidence (e), and degree of polarization (f).

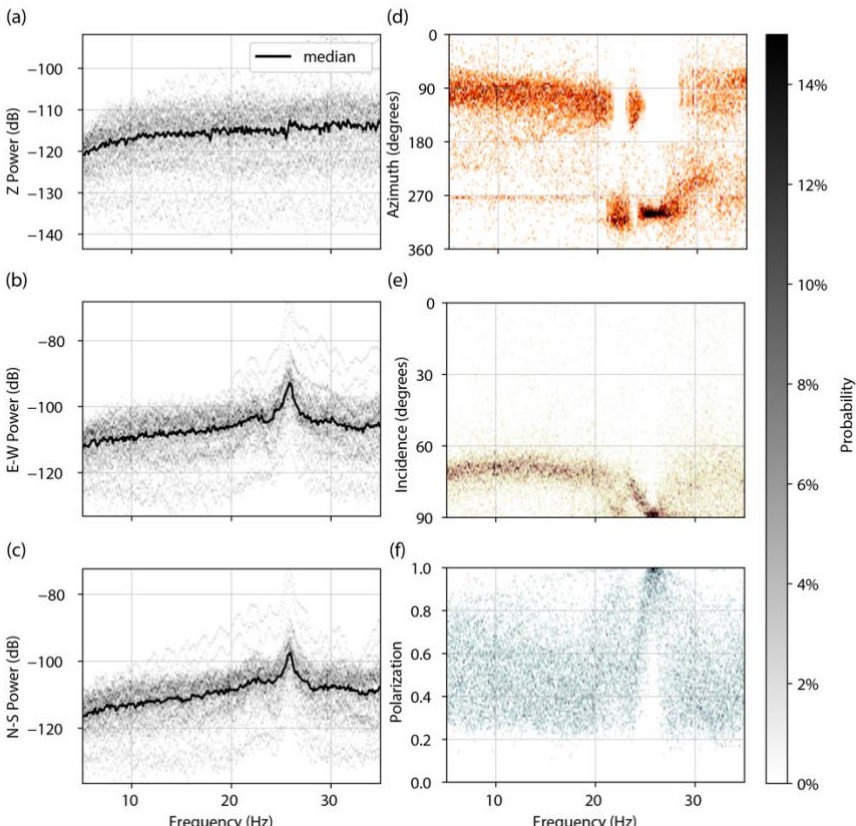

**Figure A3.** Little Egypt 2. PSDs for vertical (a), east-west (b), and north-south (c) components. Decibel powers are relative
to 1 $m^2$ $s^{-4}$ $Hz^{-1}$. Polarization attributes of azimuth (d), incidence (e), and degree of polarization (f).

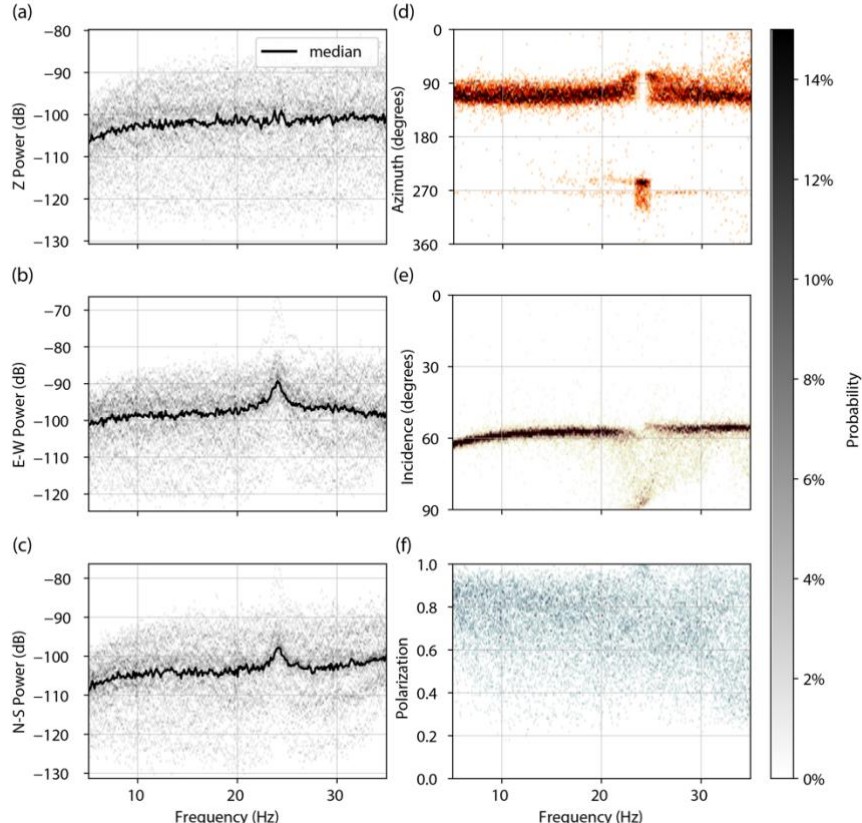

**Figure A4.** Little Egypt 4. PSDs for vertical (a), east-west (b), and north-south (c) components. Decibel powers are relative to 1 $m^2$ $s^{-4}$ $Hz^{-1}$. Polarization attributes of azimuth (d), incidence (e), and degree of polarization (f).

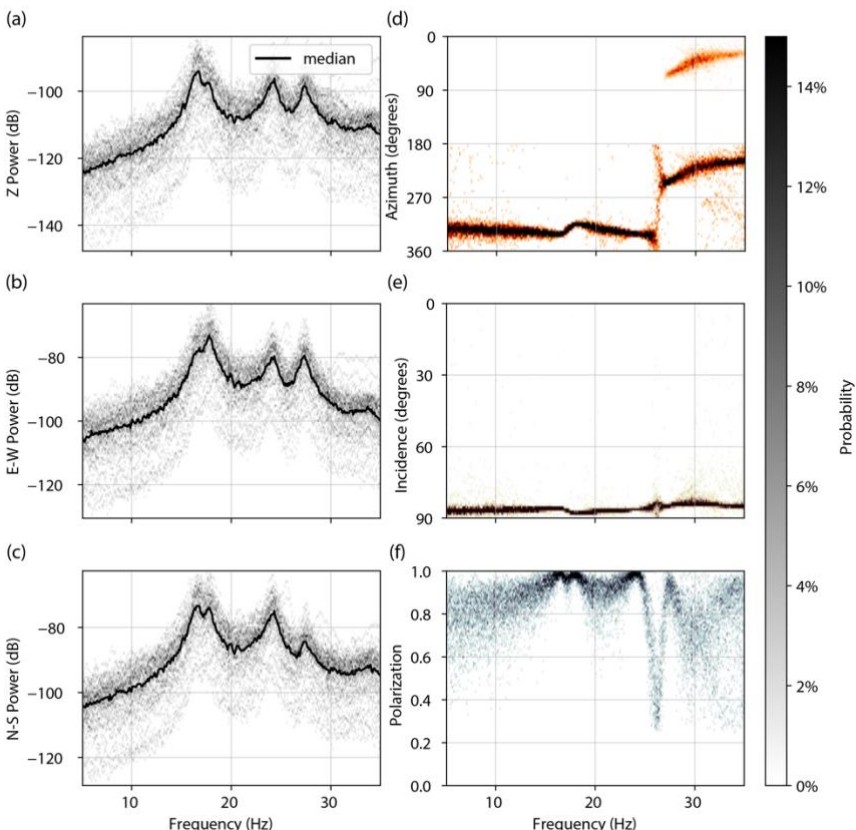

**Figure A5.** Little Egypt A. PSDs for vertical (a), east-west (b), and north-south (c) components. Decibel powers are relative to 1 m$^2$ s$^{-4}$ Hz$^{-1}$. Polarization attributes of azimuth (d), incidence (e), and degree of polarization (f).

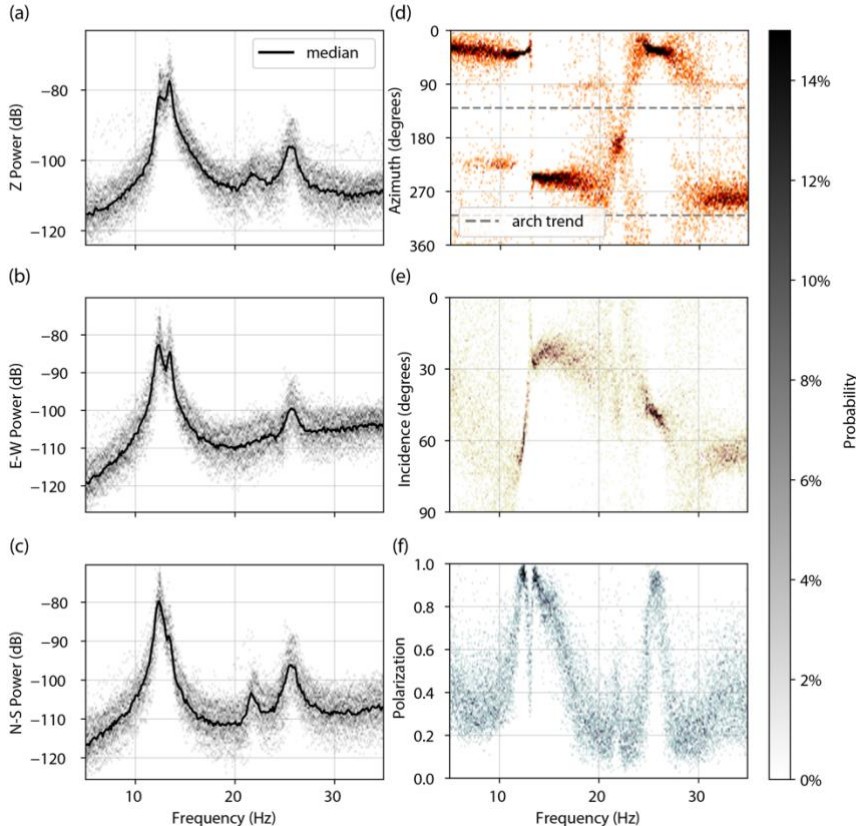

**Figure A6.** Squint Arch. PSDs for vertical (a), east-west (b), and north-south (c) components. Decibel powers are relative to 1 m$^2$ s$^{-4}$ Hz$^{-1}$. Polarization attributes of azimuth (d), incidence (e), and degree of polarization (f).

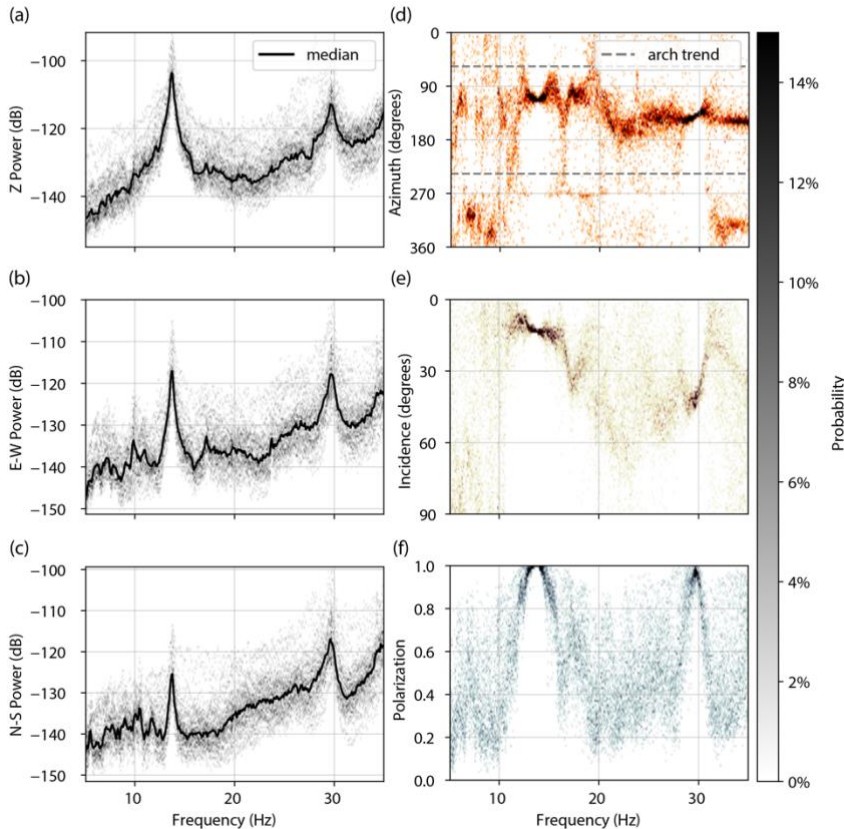


**Figure A7.** Two Bridge. PSDs for vertical (a), east-west (b), and north-south (c) components. Decibel powers are relative to 1 m$^2$ s$^{-4}$ Hz$^{-1}$. Polarization attributes of azimuth (d), incidence (e), and degree of polarization (f).

**Appendix B**

(a)  Little Egypt 2: Mode 1

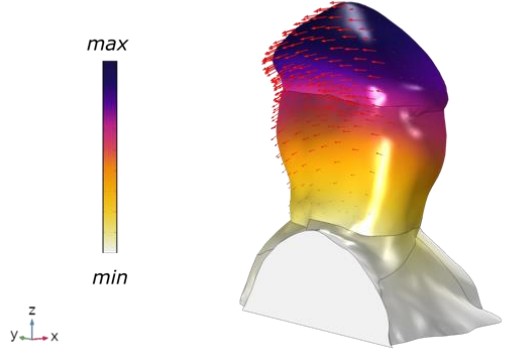

(b)  Two Bridge: Mode 1

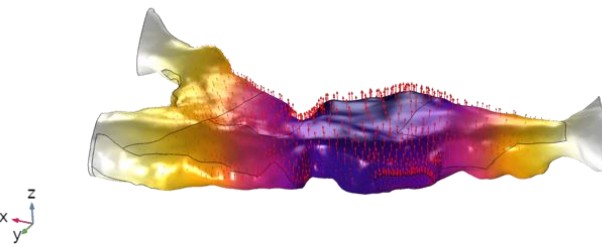

(c)  Arsenic Arch: Mode 4

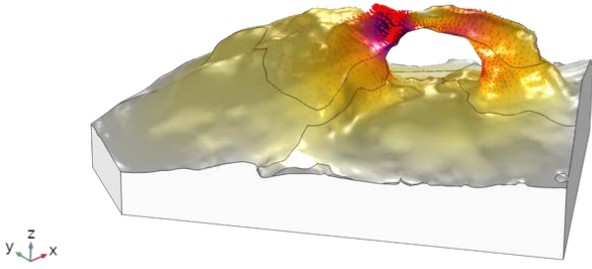

**Figure B1.** Example modeled mode shapes for Little Egypt 2 (a), Two Bridge (b), and Arsenic Arch (c).

## Appendix C: Doppler Shift

Doppler shift describes how the frequency of emitted light, sound, or another wave type from a moving object is altered by the source's motion relative to a stationary observer (Doppler, 1842). As a source approaches an observer, the wave becomes effectively compressed, raising the frequency, and as a source moves away from an observer, the opposite happens and the frequency is lowered. Following methods similar to Eibl et. al (2015), Eibl, et. al (2017), and Meng and Ben-Zion (2018), we analyzed the frequency shift of the helicopter-sourced infrasound from our field measurements. The observed frequency of a straight-line moving point source as a function of time is given in Eq. (1):

$$f_o(t) = \frac{c \cdot f_s}{c + \frac{v_s^2 \cdot (t - t_0)^2}{\sqrt{v_s^2 \cdot (t - t_0)^2 + l^2}}}, \tag{1}$$

where $f_o(t)$ is the frequency as recorded by the stationary observer, $c$ is the speed of sound in air at 20°C (343 m s$^{-1}$), and $f_s$ is the source frequency recorded at standstill. For a helicopter's blade pass frequency, $f_s$ is equal to the revolution rate of the main rotor multiplied by the number of blades. $v_s$ is the speed of the source, $t_0$ is the time at which the source is closest to the observer, and $l$ is the corresponding shortest distance.

We used parameters from the GPS track of the helicopter to calculate the Doppler shift of infrasound over time and demonstrate fit between the Doppler shift equation prediction and field measurements. We track frequency shift over time using the infrasound spectrogram: we select the frequency with the highest sound pressure level in each time window (3 s long with 90% overlap and smoothed every six points), and compare this to the resulting frequency prediction. We also fit the measured frequency shift using parameters that deliver the least amount of error, which vary slightly from the GPS parameters.

For infrasound recorded at the Squint Arch flight (Fig. C1), we use GPS parameters $v_s$ = 52 m s$^{-1}$, $t_0$ = 29 s, and $l$ = 172 m to calculate $f_o(t)$, giving a mean squared error of 8.5%. This deviation is likely caused by error from the handheld GPS. Using the best-fit parameters of $v_s$ = 55 m s$^{-1}$, $t_0$ = 28 s, and $l$ = 135 m to calculate $f_o(t)$, we achieve a mean squared error of 1.7%.

Given the maximum speed of a Bell 206 helicopter (58 m s$^{-1}$) and the absolute minimum distance the helicopter rotor can be to an observer (4 m = 1 m above the ground + 3 m helicopter height), we find the 13 Hz blade pass frequency can be shifted between 11.1 Hz and 15.6 Hz. Similarly, the 26 Hz overtone of helicopter sound energy can be shifted between 22.2 Hz and

31.3 Hz. These represent a ±15–20% shift from the standstill frequency, demonstrating the range of landform natural frequencies possibly excited by helicopter-sourced infrasound.

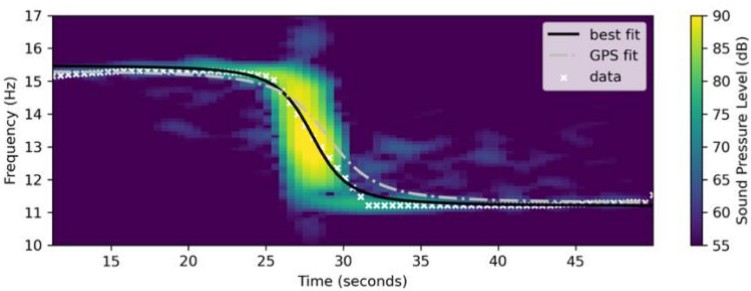

**Figure C1** Doppler shift of helicopter-sourced infrasound from a Bell 206 helicopter flying above Squint Arch. Decibel amplitudes are relative to 20E-6 Pa Hz$^{-1}$.

**Appendix D**

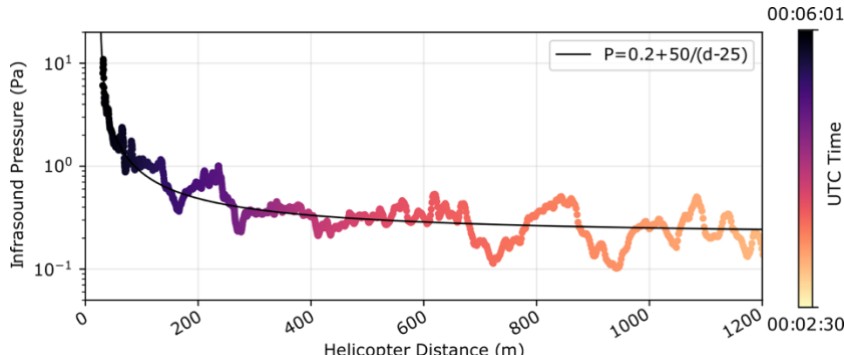

**Figure D1** Inverse relationship between helicopter-sourced infrasound pressure and distance from a Bell 206 helicopter

flying above Squint Arch. Data are filtered between 2 and 20 Hz.

**Appendix E**

**Table E1** Summary of admittance values for each study site. Arch elevation and bandpass filter range provided, along with admittance averaged for each maneuver. Times for each maneuver are also listed with the direction and elevation of the helicopter relative to the landform.

| Site | Arch Elevation | Bandpass filter | Admittance (mm/s/Pa) | Time (UTC) | Maneuver Type | Direction | Helicopter Elevation (m) |
|---|---|---|---|---|---|---|---|
| Arsenic Arch | 1463 | 22-30 Hz | 0.092 | 05 June 2019, 20:32:25 | hover | SW | 4 |
| | | | 0.079 | 05 June 2019, 20:32:58 | hover | SW | 1 |
| | | | 0.081 | 05 June 2019, 20:38:02 | circle | NE, CCW | 13 |
| | | | 0.043 | 05 June 2019, 20:34:30 | flyover | W → E | 17 |
| | | | 0.049 | 05 June 2019, 20:35:30 | flyover | W → E | 18 |
| Big Arrowhead Arch | 1400 | 10-17 Hz | 0.030 | 05 June 2019, 20:55:22 | hover | W | 22 |
| | | | 0.041 | 05 June 2019, 20:57:01 | flyover | N → S | 29 |
| | | | 0.076 | 05 June 2019, 20:57:34 | flyover | S → N | 24 |
| | | | 0.041 | 05 June 2019, 20:58:08 | flyover | N → S | 24 |
| | | | 0.049 | 05 June 2019, 20:58:40 | flyover | S → N | 20 |
| | | | 0.042 | 05 June 2019, 20:59:04 | flyover | E → W | 46 |
| | | | 0.043 | 05 June 2019, 21:04:23 | flyover | W → E | 27 |
| | | | 0.048 | 05 June 2019, 21:05:26 | flyover | W → E | 40 |
| | | | 0.039 | 05 June 2019, 21:05:48 | flyover | E → W | 53 |
| | | | 0.048 | 05 June 2019, 21:06:34 | circle | W, CCW | 41 |
| | | | 0.041 | 05 June 2019, 21:06:56 | circle | W, CCW | 46 |
| | | | 0.066 | 05 June 2019, 21:07:16 | circle | W, CCW | 46 |
| | | | 0.053 | 05 June 2019, 21:07:36 | circle | W, CCW | 49 |
| Little Egypt 2 | 1495 | 22-30 Hz | 0.056 | 05 June 2019, 20:02:25 | N/A | N/A | N/A |
| | | | 0.069 | 05 June 2019, 20:03:11 | N/A | N/A | N/A |
| | | | 0.047 | 05 June 2019, 20:03:38 | N/A | N/A | N/A |
| | | | 0.104 | 05 June 2019, 20:03:43 | hover | S | 0 |
| | | | 0.073 | 05 June 2019, 20:04:02 | hover | S | 0 |
| | | | 0.108 | 05 June 2019, 21:24:27 | hover | S | 0 |
| | | | 0.081 | 05 June 2019, 21:28:38 | hover | S | N/A |
| | | | 0.064 | 05 June 2019, 21:28:52 | hover | S | N/A |
| | | | 0.031 | 05 June 2019, 20:26:56 | flyover | NE → SW | 37 |
| | | | 0.022 | 05 June 2019, 20:27:20 | flyover | SW → NE | 48 |
| | | | 0.036 | 05 June 2019, 20:28:01 | flyover | NE → SW | 28 |
| | | | 0.021 | 05 June 2019, 20:28:35 | flyover | SW → NE | 23 |
| Little Egypt 4 | 1495 | 10-17 Hz | 0.036 | 05 June 2019, 20:03:17 | N/A | N/A | N/A |
| | | | 0.037 | 05 June 2019, 20:03:43 | hover | S | 0 |
| | | | 0.029 | 05 June 2019, 20:04:18 | hover | S | 0 |
| | | | 0.039 | 05 June 2019, 21:28:16 | hover | S | N/A |
| | | | 0.032 | 05 June 2019, 20:26:56 | flyover | NE → SW | 37 |
| | | | 0.014 | 05 June 2019, 20:27:25 | flyover | SW → NE | 31 |
| | | | 0.035 | 05 June 2019, 20:27:54 | flyover | NE → SW | 23 |
| | | | 0.014 | 05 June 2019, 20:28:28 | flyover | SW → NE | 28 |
| Little Egypt A | 1495 | 22-30 Hz | 0.106 | 05 June 2019, 20:02:30 | N/A | N/A | N/A |
| | | | 0.061 | 05 June 2019, 20:03:17 | N/A | N/A | N/A |
| | | | 0.096 | 05 June 2019, 20:03:43 | hover | S | 0 |
| | | | 0.091 | 05 June 2019, 20:04:02 | hover | S | 0 |
| | | | 0.067 | 05 June 2019, 20:04:16 | hover | S | 0 |
| | | | 0.067 | 05 June 2019, 21:24:35 | hover | S | 0 |
| | | | 0.061 | 05 June 2019, 21:28:15 | hover | S | N/A |
| | | | 0.023 | 05 June 2019, 20:26:48 | flyover | NE → SW | 20 |
| | | | 0.009 | 05 June 2019, 20:27:27 | flyover | SW → NE | 22 |
| | | | 0.027 | 05 June 2019, 20:27:56 | flyover | NE → SW | 16 |
| | | | 0.008 | 05 June 2019, 20:28:29 | flyover | SW → NE | 27 |
| Squint Arch | 1620 | 10-17 Hz | 0.047 | 01 May 2018, 00:04:09 | hover approach | W | 30 |
| | | | 0.040 | 01 May 2018, 00:04:47 | hover approach | W | 16 |
| | | | 0.037 | 01 May 2018, 00:05:21 | hover approach | W | 6 |
| | | | 0.019 | 30 April 2018, 23:46:28 | flyover | E → W | 52 |
| | | | 0.005 | 30 April 2018, 23:50:58 | distant turn around | W | 143 |
| | | | 0.018 | 30 April 2018, 23:51:56 | flyover | W → E | 62 |
| | | | 0.006 | 30 April 2018, 23:53:02 | distant turn around | E | 85 |
| | | | 0.020 | 30 April 2018, 23:53:55 | flyover | E → W | 39 |
| | | | 0.030 | 30 April 2018, 23:55:47 | flyover | W → E | 27 |
| | | | 0.023 | 30 April 2018, 23:57:15 | flyover | E → W | 35 |
| | | | 0.030 | 01 May 2018, 00:01:04 | flyover | SE → NW | 52 |
| Two Bridge | 2337 | 10-17 Hz | 0.040 | 26 October 2017, 16:30:05 | flyover | S → N | 505 |
| | | | 0.062 | 26 October 2017, 15:55:48 | N/A | N/A | N/A |

## Data Availability

Data from this study are available at https://doi.org/10.7278/S50D-41SW-THMA.

## Author Contribution Statement

The manuscript was written by RF with significant contributions from all co-authors. RF, PG, and JM acquired field data. RF carried out data processing with contributions from PG and JM. All authors reviewed and approved the manuscript.

## Competing Interests

The authors declare that they have no conflict of interest.

## Acknowledgements

This study was funded by the National Science Foundation grant EAR-1831283, the University of Utah Vice President for Research, and by the Utah Science Technology and Research Initiative. We thank the Native American Consultation Committee advising Rainbow Bridge National Monument, and the Hopi, Navajo, Southern Paiute, Ute, and Zuni members therein, along with the National Park Service, for stimulating this study. Jeffrey Johnson provided invaluable support for infrasound data acquisition. Erin Bessette-Kirton, Cynthia Gardner, Ammon Hatch, Holly Hurtado, Brendon Quirk, Anna Stanczyk, Kathryn Vollinger, and Holly Walker helped with field work. We also thank Michael Tsesarsky and an anonymous reviewer for helpful feedback. Data from this study are available at https://doi.org/10.7278/S50D-41SW-THMA. The authors declare no financial conflicts of interest.

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
