# Peer review of "Vibration of Natural Rock Arches and Towers Excited by Helicopter-Sourced Infrasound"

_Earth Surface Dynamics, 2021_

## Author Comment (AC1)

Dear Editors and Reviewers,

We appreciate the time you have spent towards the careful reading and reviewing of our manuscript. We have reviewed the comments and updated the manuscript accordingly. We believe the feedback has improved the paper and hope it is satisfactory for publication; thank you for your contributions.

On behalf of all authors,
Riley Finnegan

Reviewer #1

I have read the paper with interest. The article is well written and reads well. It is novel and presents a new approach towards man-made load on natural landforms of cultural and esthetic value. The research methodology is sound and clearly presented.

My main critique is towards the unneeded extrapolations and conjectures. The results speak for themselves, even if their practical significance is of second order. For example, lines 233-237, "Thus the range of landforms susceptible to high-amplitude resonance from helicopter-sourced infrasound is likely narrow, i.e., the conditions such as frequency and modal vector alignment limit the number of landforms that might experience large vibration amplitudes during helicopter flight. However, a single past study reported peak vibration velocities of 4.1 mm s-1 for a ~13 m tall rock pinnacle (with similar values for two neighboring pinnacles), demonstrating the viability of these elevated amplitudes (King, 2001). Followed by lines 238 – 239, "We additionally note that heavier military helicopters not studied here likely generate higher power infrasound than lighter civilian models (Hanson et al., 1991), and further study is needed to test the effects of military helicopter overflights on the vibration response of rock landforms.

What can we learn from a single pass? For example, are military helicopters allowed to fly and hover over National Parks and Monuments?

**Response:** We've removed the word "single" from this line, but note that we were referring to a *past* study, not a "single pass". We think it's important to keep the note about military helicopters as they do fly over the parks, other public lands, and many places around the world.

Lines 246 – 248: "While this scenario is limited, there are thousands of arches, towers, and other culturally significant valuable rock landforms in Utah alone (e.g., Stevens and McCarrick, 1988), and a multitude of others other nationally significant geologic features around the world".

This is stating the obvious. Instead, please provide a range of height, span, stiffness, or any other mechanical attribute of landforms that would be affected by helicopter-generated infrasound pressure.

**Response:** We updated the text to provide examples of arches and places with towers and hoodoos that have or are likely to have natural frequencies that are within the range of Doppler shifted helicopter infrasound.

My suggestion is to avoid generalized statements where and when possible.

Also, there are some minor remarks:

1. Table 1 and Figure 2. There seems to be a mismatch with the frequencies of Arsenic and Squint arches. In the table, there are three frequencies for the Squint arch and a single frequency for Arsenic. The figure shows opposite.

**Response:** Assuming it was Figure 1 that was meant to be referenced and not Figure 2, we have updated the figure to include a shaded region of the Doppler shifted range of the fundamental and first overtone of helicopter infrasound behind each landform spectra. This hopefully conveys the alignment between the natural frequencies of interest--for instance, while Arsenic Arch has 4 natural frequencies shown, only the 25.6 Hz frequency aligns with the Doppler shifted helicopter infrasound, and Squint Arch's 12.5, 13.8, and 26.0 Hz natural frequencies align with the Doppler shifted infrasound. We also added a figure in the appendix detailing the full polarization analysis for each landform.

2. Table 1. Young's modulus of what? Structure, rock mass, rock?

**Response:** Young's modulus of the rock mass. We updated the table caption.

3. Line 78, please provide the models of broadband and geophones.

**Response:** We updated the sentence to include the models.

4. Table 2. I am not convinced it is necessary.

**Response:** Our literature review leads us to believe this information is neither widely known nor available, and the table provides information on infrasound produced by helicopter models other than the one used in our study. Therefore, we prefer to keep it in the main text. Additionally, this table serves as a quick reference that accompanies this work for others to draw upon to translate this study to their area or problem.

5. Figure 2e is in mm/sec units; it is hard to see the "…increased 100–1000 times during helicopter flight…" Please provide numbers for median background noise.

**Response:** We provide arch vibration power in Figure 2f in order to see the measured increase in a logarithmic scale.

6. Line 223. It would be of value to provide a correlation between landform(s) vibration values and the expected change of stresses.

**Response:** We updated the text to provide an approximation of the additional stress on the arch using a fixed-fixed beam analytical approximation analogous to Squint Arch.

Reviewer #2

The authors present an interesting study associated with the vibration of rock arches and towers caused by helicopter-sourced infrasound. Both seismic and infrasound observations are carefully analyzed to evaluate the response of natural landforms to anthropogenic sources. The pieces of evidence and

analyses are well organized and the paper is well written. However, there are a few aspects that need a further demonstration.

First, the authors mentioned 3D models of landforms were built to estimate the eigenfrequencies of the arches and towers using finite-element simulations. This is an important component of the analysis and can provide insightful discussion on this topic. Please consider, at least, adding a dedicated paragraph describing the details.

**Response:** We added text to describe how we developed the 3D models and the extent of their use in this study, along with a new appendix figure showing modal deformation fields of some landforms featured in our study.

In addition, in L80-84, a couple of procedures and methods provided by the previous study have been cited without sufficient details. It is difficult for the audience to understand what has been done. Please add a sentence or two to briefly summarize these methods and procedures.

**Response:** We added details to the methods section about the polarization analysis, along with detailed polarization analysis figures

Further, in section 5.3, the author compared the maximum vibration velocities measured on landforms with the empirical levels of previous studies. Using vibration velocity is only a practical way to evaluate the damage. The damage should be determined by the stress perturbations and the structural strength. As the 3D finite-element model are available, it will be great to estimate the maximum stress/strain perturbations inside the landforms caused by the infrasound loading of helicopters. What's the order of the stress perturbation when loading on a landform of ~10 Pa at its natural frequencies? The stress perturbation is a more direct parameter to compare with the rock strength. This should be the highlight of this paper and will cause an impact.

**Response:** As it is beyond the scope of this study to perform a fluid mechanics simulation to properly model the interaction between a 3D model of an arch and helicopter infrasound, and because doing so would introduce a new set of uncertainties and assumptions, we approximated the additional stress on the arch using a fixed-fixed beam analogous to Squint Arch and updated the text with these approximations.

Last but not the least, when introducing new parameters, e.g., admittance, it is always good to demonstrate what does it mean when the value is high and low.

**Response:** We updated the text to provide additional meaning.

Here are the detailed comments:

Figure 1. Can the authors also show the spectra of the infrasound records for comparison?

**Response:** Figure 1a shows the helicopter infrasound spectra. We have additionally updated the figure to include a shaded region of the Doppler shifted range of the fundamental and first overtone of helicopter infrasound behind each landform spectra.

Table 2. Are the overtones always the multiples of blade pass frequency?

**Response:** Yes, we state this in line 133.

L80-81: Please briefly summarize the method performed to estimate the natural frequencies.

**Response:** Text is updated, see above response to general comment.

L82-85: Please consider adding a supplemental figure comparing the observed and modeled nature frequencies of the 4 arches and 3 towers analyzed in this study.

**Response:** The arches and comparison between measured/modeled results have already been described by Geimer et al. (2020). As the match or comparison between observed and modeled natural modes is not required for this study, we find it more valuable to highlight the spectral details of the dataset, which serve as the basis for our modal analysis. These are found in a new appendix. We additionally included examples of excited modes from our 3D models for three different landforms in a separate figure in the aforementioned appendix.

L97-99: Did the authors perform lowpass filtering before downsampling the records? Otherwise, there will be artifacts associated with aliasing.

**Response:** Yes, all data are lowpassed prior to downsampling.

L105: The authors mentioned the orientation of maximum vibration for arches is in the vertical. Can this be confirmed by the seismic observation? Please add a sentence or two to discuss.

**Response:** Yes, and we have now added figures in the appendix with the polarization analysis of the seismic data detailing the spectral attributes for each landform. These are also described in detail by Geimer et al. (2020).

L106: Are the dominant polarization azimuth associated with the geometry of hoodoos?

**Response:** Two of the three hoodoos are nearly cylindrical in cross-section and therefore there is no strong preference dictated by geometry.

L108: "We plotted …" Is this referring to Figure 2 or Figure 3?

**Response:** We updated the line to refer to the correct figure, Figure 3f.